

# Resolving multiple geological events using in situ Rb-Sr geochronology: implications for metallogenesis at Tropicana, Western Australia

Hugo K. H. Olierook[1,2,3]*, Kai Rankenburg[1,3], Stanislav Ulrich[4], Christopher L. Kirkland[1,2], Noreen J. Evans[1,3], Stephen Brown[4], Brent I. McInnes[3], Alexander Prent[1,3], Jack Gillespie[1], Bradley McDonald[1], and Miles Darragh[4]

[1]School of Earth and Planetary Sciences, Curtin University, GPO Box U1987, Perth, WA 6845, Australia
[2]Centre for Exploration Targeting – Curtin Node, Curtin University, GPO Box U1987, Perth, WA 6845, Australia
[3]John de Laeter Centre, Curtin University, GPO Box U1987, Perth, WA 6845, Australia
[4]AngloGold Ashanti Australia Ltd., 140 St. Georges Terrace, Perth, WA 6000, Australia

*Correspondence to*: Hugo K. H. Olierook ([hugo.olierook@curtin.edu.au](mailto:hugo.olierook@curtin.edu.au))

**Abstract.** Dating multiple geological events in single samples using thermochronology and geochronology is relatively common but it is only with the recent advent of triple quadrupole LA-ICP-MS that in situ Rb-Sr dating has become a more commonly applied and powerful tool to date K- and Rb-bearing minerals. Here, we date, for the first time, two generations of mineral assemblages in individual thin sections using the *in situ* Rb-Sr method. Two distinct mineral assemblages, both probably associated with Au mineralization, are identified in samples from the Tropicana gold mine in the Albany–Fraser Orogen, Western Australia. For Rb-Sr purposes, the key dateable minerals are two generations of biotite, and additional phengite associated with the second assemblage. Our results reveal that the first, coarse-grained generation of biotite grains records a minimum age of 2535 ± 18 Ma, coeval with previous $^{40}Ar/^{39}Ar$ biotite, Re-Os pyrite and U-Pb rutile results. The second, fine-grained and recrystallized generation of biotite grains record an age of 1207 ± 12 Ma across all samples. Phengite and muscovite yielded broadly similar results at ca. 1.2 Ga but data is overdispersed for a single coeval population of phengite and shows elevated age uncertainties for muscovite. We propose that the ca. 2530 Ma age recorded by various geochronometers represents cooling and exhumation, and that the age of ca. 1210 Ma is related to major shearing associated with the regional deformation associated with Stage II of the Albany–Fraser Orogeny. This is the first time that an age of ca. 1210 Ma has been identified in the Tropicana Zone, which may have ramifications for constraining the timing of mineralization in the region. The *in situ* Rb-Sr technique is currently the only tool capable of resolving both geological events in these rocks.



## 1.0 Introduction

The ability to date multiple events in individual samples has important consequences for developing a comprehensive understanding of the geological history of complex terranes. The U-Pb method has long been employed to date crystallization, metamorphism and hydrothermal events, often by targeting cores and rims in individual grains. Many U-bearing minerals have recorded multiple ages due to their ability to participate in metamorphic/hydrothermal reactions or become (partially) reset by

events above mineral closure temperatures, including zircon (Liu et al., 2012), monazite (Rasmussen et al., 2007), titanite (Olierook et al., 2019), rutile (Zack and Kooijman, 2017) and apatite (Kirkland et al., 2018). However, not all geological events are associated with (partial) reset or new growth of U-bearing minerals. In these scenarios, it is important to examine alternative minerals that may provide a more complete record of the geological history.

The Rb-Sr isotopic system is particularly valuable for geochronology, as Rb is sufficiently abundant in common K-bearing minerals like biotite, muscovite and K-feldspar that are abundant in a wide variety of rocks, and are readily mobilized during fluid–rock interactions (Riley and Compston, 1962;Attendorn and Bowen, 1997). $^{87}$Sr decays to $^{87}$Rb with a recently revised decay constant of $1.3972 \pm 0.0045 \times 10^{-11}$ a (equivalent to half-life of ~49.6 Ga; Villa et al., 2015). However, the most significant disadvantage of traditional Rb-Sr geochronology is the inability to perform *in situ* dating via secondary ion mass

spectrometry (SIMS) or laser ablation inductively coupled plasma mass spectrometry (LA-ICP-MS; Nebel, 2013). Although several studies have dated mineral separates on a small scale (Glodny et al., 2002;Glodny et al., 2003), some even texturally-constrained by micromilling (Chen et al., 1996;Charlier et al., 2006), the Rb-Sr technique could not compete with the <100 μm diameter resolution of the U-Pb method.

The major obstacle with *in situ* Rb-Sr geochronology is the isobaric interference of different isotopes, most notably that of $^{87}$Rb and $^{87}$Sr (Zack and Hogmalm, 2016). Pioneering work from Moens et al. (2001) and Vanhaecke et al. (2003) showed that it was possible to achieve chemical separation of interfering $^{87}$Rb from $^{87}$Sr inside a conventional ICP-MS by directing the ion beam through a dynamic reaction cell with $CH_3F$ gas to produce $SrF^+$ ($m/v \approx 106$) but leave Rb unaffected ($m/v \approx 87$). However, this technique was relatively imprecise ($\pm$ ~10%; Vanhaecke et al., 2003), particularly when compared to *in situ* U-Pb methods

($< \pm 2\%$) and still required dissolution of the sample.

With the recent advent of 'triple quadrupole' LA-ICP-MS, it is now possible to perform *in situ* Rb-Sr dating at precision that that rivals *in situ* U-Pb geochronology (Zack and Hogmalm, 2016;Hogmalm et al., 2017). A reaction cell located between two quadrupoles is filled with a selected gas (e.g., $N_2O$, $SF_6$, $O_2$) that reacts with $Sr^+$ ions but leaves $Rb^+$ unaffected. Thus, the first

quadrupole is used to filter ions of a specific mass (e.g., $^{87}$Rb and $^{87}$Sr) to enter the reaction cell and the second quadrupole separates the $^{87}$Rb from the reacted (mass-shifted) Sr (e.g., $^{87}Sr^{16}O$; now $m/v$ 103, see supplementary Fig. A for graphic illustration of this process).



Following the work of Zack and Hogmalm (2016) and Hogmalm et al. (2017) for assessing the most suitable reaction cell
gases, several publications have attempted to solve geological problems using the *in situ* Rb-Sr technique (Şengün et al.,
2019;Tillberg et al., 2017;Tillberg et al., 2020). All these studies, except the one from Tillberg et al. (2020), identified only a
single age population within individual samples, which could have been resolved (at higher precision) with solution Rb–Sr or
$^{40}Ar/^{39}Ar$. Tillberg et al. (2020) observed multiple age populations in their samples, but these were from mineral separates and
the textural context was not preserved. To date, no published study has taken full advantage of the spatial resolving power of
the *in situ* Rb-Sr technique whilst retaining textural context.

Here, we analyzed in thin sections, the *in situ* Rb-Sr ages of two mineral assemblages developed in distinctly different
deformation microstructures in the Tropicana Zone of the Albany–Fraser Orogen, southwestern Australia. For Rb-Sr purposes,
we date (i) biotite from both assemblages, (ii) apatite from both generations, (iii) phengite from assemblage 2, and (iv)
muscovite from assemblage 2. Ultimately, this work demonstrates the use of coupled *in situ* Rb-Sr geochronology and
microstructural analysis for identifying and resolving multiple geological events in individual samples.

## 2.0  Geological Background

### 2.1  Geological history of the Albany–Fraser Orogen

The Albany-Fraser Orogen is a Proterozoic orogenic belt that girdles ~1200 km of the south and southeastern margins of the
Archean Yilgarn Craton in Western Australia. This belt had a protracted Proterozoic history that included a series of
extensional and compressional events at ca. 2720–2530 Ma, 1810–1650 Ma and 1330–1140 Ma (Spaggiari et al., 2015). The
Albany-Fraser Orogen comprises several lithotectonic domains including the Northern Foreland, Tropicana Zone, Biranup
Zone, Nornalup Zone and Fraser Zone (**Fig. 1**), and principally represents the reworked margin of the Archean Yilgarn Craton
(Kirkland et al., 2011). Each zone comprises minor to dominant components of Archean heritage variably reworked by
Paleoproterozoic and Mesoproterozoic tectonomagmatic events.

The earliest event in the belt at ca. 2720–2530 Ma was restricted to the Tropicana Zone (see section 2.2) and was followed by
magmatism from 1.81 Ga to 1.65 Ga in the Tropicana, Biranup and Nornalup zones (Smithies et al., 2015). This earlier
Paleoproterozoic magmatism is divided into three pulses: Salmon Gums Event (1.81–1.80 Ga), Ngadju Event (1.77–1.75 Ga)
and Biranup Orogeny (1.70–1.65 Ga; Kirkland et al., 2011;Spaggiari et al., 2015;Smithies et al., 2015). The tectonic setting in
which this significant Paleoproterozoic magmatism occurred is not well constrained, however, it is generally interpreted to
represent an extensional event (Spaggiari et al., 2015;Hartnady et al., 2019;Smits et al., 2014), with short-lived pulses of
compression (i.e. Zanthus Event; Kirkland et al., 2011;Smithies et al., 2015).





The majority of the magmatism in the Albany-Fraser Orogen occurred during arc-accretion and subsequent reworking at 1330

Ma and 1200 Ma, respectively (Spaggiari et al., 2015). The Albany-Fraser Orogen shares a heritage with Wilkes Land in East Antarctica, and these two orogenic belts were contiguous during the late Mesoproterozoic as a result of Rodinia assembly (Morrissey et al., 2017;Clark et al., 2000). Stage I of the Albany-Fraser Orogeny (1330–1260 Ma) was a widespread high-temperature, moderate- to high-pressure event accompanied by felsic and mafic magmatism (Clark et al., 2014). Stage I is generally interpreted as the collision between the Western Australian and Mawson Craton (Clark et al., 2000;Bodorkos and

Clark, 2004). Stage II of the Albany-Fraser Orogeny is considered to reflect intracratonic orogenesis (Spaggiari et al., 2009;Spaggiari et al., 2015;Spaggiari et al., 2014). This stage is associated with craton-verging thrusting, high-temperature and moderate-pressure metamorphism, and mainly felsic magmatism at ca. 1225–1140 Ma (Dawson et al., 2003;Nelson et al., 1995). Mafic intrusions associated with Stage II are not known in the eastern Albany-Fraser Orogen but have recently been documented at $1134 \pm 9$ (U-Pb zircon) and $1131 \pm 16$ Ma (U-Pb baddeleyite) in the Bunger Hills, Wilkes Land (Stark et al.,

105 2018).

## 2.2 Geological and mineralization history of the Tropicana Zone

The Tropicana Zone is located along the northeastern margin of the Yilgarn Craton (Fig. 1). Seismic sections across the Tropicana Zone reveal a northwest directed, imbricate thrust stack formed in a foreland setting by thrusting of the Tropicana Zone up along a major thrust surface known as the Plumridge Detachment (Occhipinti et al., 2014;Occhipinti et al., 2018).

This thrust transported the Tropicana Zone onto the Yamarna Terrane of the Yilgarn Craton (Occhipinti et al., 2018).

The Tropicana Zone includes the Tropicana gold mine and several prospects to the northeast and southwest (Fig. 1; Occhipinti et al., 2018;Spaggiari et al., 2014). A moderately foliated metagranite (Hercules Gneiss) sampled close to the Tropicana gold mine yielded a U-Pb age of $2722 \pm 15$ Ma on oscillatory-zoned zircon cores interpreted to represent the magmatic

crystallization age of the granite (Kirkland et al., 2015). A younger age of $2640 \pm 10$ Ma on zircon rims from the same sample was interpreted as the age of a high-grade metamorphic overprint in the zone. In the Tropicana gold mine itself, a similar minimum age of crystallization ($2638 \pm 4$ Ma) was acquired from the syenitic lithofacies of the Tropicana Gneiss (Doyle et al., 2015). The Hercules Gneiss has broadly dioritic compositions and a very narrow range of low $SiO_2$ (58.1–63.6 wt%), and is classified as a sanukitoid (Kirkland et al., 2015). Sanukitoid magmas, usually are produced from metasomatized mantle in

an arc setting (Martin et al., 2005), are known for gold fertility and are interpreted as a likely source of gold in the Tropicana Zone, although the gold may have been remobilised on several occasions (Kirkland et al., 2015). Sanukitoid intrusions commenced at $2692 \pm 16$ Ma near the start of a prolonged mid-amphibolite to lower granulite facies metamorphism in the Tropicana Zone that persisted until ca. 2530 Ma (Kirkland et al., 2015;Doyle et al., 2015). Kirkland et al. (2015) interpreted the characteristic high-grade metamorphic textures and grain shapes of zircon as evidence of a prolonged period of granulite

facies metamorphism (Atlantis Event) that formed many of the gneisses in the Tropicana Zone. Structural and isotopic data



imply that the Tropicana Zone was held at a deep-crustal level during much of the Neoarchean (Tyler et al., 2015;Occhipinti et al., 2018).

Exhumation and retrogression to greenschist facies metamorphic conditions associated with folding and development of thrust shear zones occurred at ca. 2530 Ma (Blenkinsop and Doyle, 2014;Doyle et al.;Doyle et al., 2015). Thrusting onto the Yilgarn Craton is thought to have led to ingress of fluids and Au mineralization at ca. 2530 Ma (Occhipinti et al., 2018;Doyle et al., 2015). The age of ca. 2530 Ma from the Tropicana gold mine is constrained from a biotite $^{40}$Ar/$^{39}$Ar age of 2531 ± 14 Ma, recalculated using the decay constant of Renne et al. (2011), and an imprecise pyrite Re-Os age of ca. 2505 ± 50 Ma. Additionally, a tungsten-rich rutile population exsolved from a coarse-grained biotite yielded dates between 2539 ± 22 Ma and 2479 ± 10 Ma, overdispersed for a single population (Doyle et al., 2015). There is evidence from the work of Doyle et al. (2015) for subsequent resetting of geochronometers. Although Doyle et al. (2015) advocates for a 2521 ± 5 Ma age for rutile formation, it is more likely this represents partial resetting. Similarly, pyrite Pb/Pb results show scatter between ca. 2500 Ma and 1800 Ma, disturbed $^{40}$Ar/$^{39}$Ar spectra show a range of individual steps between ca. 2.0 and 1.8 Ga and U-Pb zircon and monazite ages show partial loss of Pb towards Mesoproterozoic ages but with poorly constrained lower intercepts (ca. 1.3–1.1 Ga). Whether these dates represent distinct events at ca. 2.4, 1.8 Ga and/or 1.3–1.1 Ga or are a continuum of dates towards younger ages remains unknown.

## 3.0 Sample selection

The sampling strategy for the Tropicana gold mine followed that of Blenkinsop and Doyle (2014), focusing on the Au-mineralized D3 shear zone. There is a natural strain gradient from undeformed syenitic gneiss host rock to transient low-strain, up to high-strain zones. A total of ten samples were selected from diamond drill cores (photos in Supplementary Fig. B) from three main pits in the Tropicana gold mine (Table 1, Figs. 1–2). All samples are perthitic K-feldspar dominated rocks with minor biotite and quartz, deformed at low strains to a brittle-ductile microstructure at greenschist facies conditions. Additional phases include albitized plagioclase, biotite, phengite, quartz, calcite/dolomite, pyrite, zircon and monazite (Supplementary Fig. B).

Three samples were selected from satellite prospects proximal to the Tropicana gold mine and within the Tropicana Zone, one each from the New Zebra, Iceberg and Angel Eyes prospects (Table 1, Figs. 1–2). The sample from the New Zebra prospect displays parasitic folding defined by muscovite and quartz (Table 1, Supplementary Fig. B). The Iceberg and Angel Eyes samples are both strongly foliated, with foliation defined by quartz, phengite ± altered plagioclase (Table 1, Supplementary Fig. B).





A subset of four of the ten samples from the Tropicana gold mine were selected for *in situ* Rb-Sr geochronology (Table 1, Figs. 1–2). From the satellite prospects, the sample from the New Zebra prospect was selected.

**4.0 Methods**

**4.1 Thin section preparation and imaging**

Standard polished thin sections prepared at Minerex Services, Esperance, Western Australia, were imaged in plane- and cross-polarized, transmitted light on an Axio II optical microscope at the School of Earth and Planetary Sciences, Curtin University. Thin sections were subsequently carbon coated and analyzed using a Tescan Integrated Mineral Analyser (TIMA) at the John
de Laeter Centre (JdLC) at Curtin University to aid in mineral identification. TIMA (a is a field emission gun scanning electron microscopy) is equipped with four electron dispersive X-ray spectrometers (EDS), capable of recording 420k X-ray counts per second. Thin sections were analyzed in 'dot-mapping' mode with a rectangular mesh at a step-size of 3 μm for backscattered electron (BSE) imaging. One thousand EDS counts are collected every $9^{th}$ step (i.e., 27 μm) or when the BSE contrast changes (i.e., a change in mineral phase). For a given mineral grain, EDS counts are integrated across the entire grain. TIMA analyses
used an accelerating voltage of 25 kV, a beam intensity of 19, a probe current of 6.74–7.01 nA, a spot size of 67–90 nm and a nominal working distance of 15 mm. After imaging and EDS collection, BSE signals and EDS peaks are referenced to a mineral library for automatic mineral classification.

Full thin section photomicrographs and TIMA images of each sample are in Supplementary Fig. B.

***In situ* Rb-Sr geochronology**

*In situ* Rb-Sr data were collected on sample thin sections in the GeoHistory Facility, JdLC, Curtin University, across three sessions. For all sessions, a RESOlution LR 193 nm ArF excimer laser with Laurin Technic S155 sample cell was used. Laser settings comprised a beam diameter of 87 μm (session 1) or 64 μm (sessions 2–3), on-sample energy of 2.5 J cm$^{-2}$, a repetition
rate of 5 Hz, 60s of analysis time and 30s of on-peak background acquisition with the laser off. All analyses were preceded by two cleaning pulses. Laser fluence was calibrated each day using a hand held energy meter, and subsequent analyses were performed in constant fluence mode. The Laurin Technic S155 sample cell was flushed with ultrahigh purity He (320 mL min$^{-1}$) with added $N_2$ (1.2 mL min$^{-1}$), both of which were passed through an inline Hg trap. High purity Ar was used as the ICP-MS carrier gas (flow rate ~1 L min$^{-1}$).

Rb-Sr data were collected on an Agilent 8900 triple quadrupole mass spectrometer in MS/MS mode (see Supplementary Table A for all relevant tuning and acquisition parameters) using $N_2O$ reaction gas following the pioneering work of Cheng et al. (2008) and Hogmalm et al. (2017). Each analytical session consisted of first tuning gas flows and ICP-MS ion lenses in single



quad mode for sensitivity and a flat mass response curve, followed by adjustment for robust plasma conditions, including

$^{238}U/^{232}Th \sim 1$, $^{206}Pb/^{238}U \sim 0.2$ and $^{238}UO/^{238}U < 0.004$ on NIST610 glass (Kent et al., 2004). The mass spectrometer was then set to MS/MS mode, and $N_2O$ was added ($\sim 0.25$ mL min; not calibrated$^{-1}$) to the reaction cell. The reaction cell was flushed with $N_2O$ for several hours before sample analysis to ensure signal stability. NIST610 was used to tune $N_2O$ to maximise intensity at mass 104 ($^{88}Sr^{16}O$), while maintaining zero counts at mass 101 ($^{85}Rb^{16}O$). Finally, pulse-analog (P/A) conversion factors for $^{88}Sr^{16}O$ (as $^{104}Pd$) and $^{87,85}Rb$ were determined in single quad mode on NIST610 reference glass and pressed powder

tablets of phlogopite Mica-Mg, respectively (Govindaraju, 1979;Kröner et al., 1996;Morteani et al., 2013;Hogmalm et al., 2017), by varying laser spot sizes and/or laser repetition rate to yield $\sim 2$ Mcps per analyte.

We designed the analytical protocol to stay below the P/A conversion thresholds for Rb and Sr by reducing ablation spot size, laser repetition rate, and /or laser energy. Maximizing count rates for $^{87,86}Sr$ implied that $^{88}Sr$ was not available for mass bias

correction. We thus followed the approach of Hogmalm et al. (2017) to calibrate $^{87}Sr/^{86}Sr$ directly against NIST610 ($^{87}Sr/^{86}Sr = 0.709699 \pm 0.000018$; Woodhead and Hergt, 2001) to calculate $^{87}Rb/^{86}Sr$ from certified values at $2.390 \pm 0.005$ (Woodhead and Hergt, 2001). In order to check for matrix sensitivity of measured Sr isotopic compositions using this approach, we interspersed a megacrystic plagioclase and a modern shark tooth (apatite) with the samples as external standards. The measured results for plagioclase ($^{87}Sr/^{86}Sr = 0.7037 \pm 0.0013$; 2SE; $n = 15$) and shark tooth apatite ($^{87}Sr/^{86}Sr = 0.7106 \pm 0.0013$; 2SE; $n$

$= 15$) are in excellent agreement with the published Sr isotopic compositions of $0.70310 \pm 0.00007$ (plagioclase Mir a; Rankenburg et al., 2004), and modern marine seawater $^{87}Sr/^{86}Sr$ of $0.709174 \pm 0.000003$ (McArthur et al., 2006), respectively, and attest to the validity of our analytical protocol. Our measured $^{87}Sr/^{86}Sr$ for mica-Mg calibrated against NIST610 over the course of this study was $1.8692 \pm 0.0022$ (2SE, $n = 28$), and we used this value along with a crystallization age of $519.4 \pm 6.5$ Ma (2$\sigma$) and initial $^{87}Sr/^{86}Sr$ of $0.72607 \pm 0.00070$ (Hogmalm et al., 2017) to calculate a mean $^{87}Rb/^{86}Sr$ for mica-Mg of 156.9

$\pm 2.3$, with all errors propagated in quadrature.

Whereas all Rb-Sr isotopic analyses were initially normalized and drift-corrected with factors determined from NIST610, an additional matrix correction to $^{87}Rb/^{86}Sr$ was only applied to biotite analyses, with uncertainties on Mica-Mg and the unknown analyses propagated in quadrature. These corrections were also applied to phengite and muscovite, but with the caveat that

Mica-Mg may not be a concentration-matched standard for these minerals. Because calculated ages from sample biotite mainly depend on accurate determination of the Rb/Sr fractionation factor, a secondary mica standard of known age is highly desirable. To this end, analyses of unknowns were additionally bracketed with in-house biotite reference material CK001B ($422 \pm 6$ Ma; Kirkland et al., 2007;Daly et al., 1991). Sample CK001B was collected by Daly et al. (1991) but not dated precisely. Collected < 50 km from CK001B and having experienced equivalent Caledonian metamorphism, the age of sample CK009 was

determined from amphibole, whole-rock and biotite Rb-Sr solution analyses (Kirkland et al., 2007). CK009 yielded an age of $422 \pm 6$ Ma ($n = 5$, MSWD = 0.57, $p = 0.68$), recalculated using the decay constant of Villa et al. (2015), and an initial $^{87}Sr/^{86}Sr$ ratio of $0.7108 \pm 0.0001$ (Kirkland et al., 2007). Repeated analytical results from sessions 1–3 on adjacent spots show no





systematic variation in Rb-Sr age (see supplementary Fig. C). During analytical sessions 1, 2 and 3, sample CK001B yielded biotite ages of $413 \pm 4$ ($n = 38$, MSWD = 1.2, $p = 0.18$), $414 \pm 5$ ($n = 38$, MSWD = 0.99, $p = 0.49$) and $429 \pm 8$ ($n = 46$, MSWD

$= 0.28$, $p = 1.00$), respectively (Supplementary Fig. C). All three sessions yielded a combined age of $416 \pm 3$ ($n = 122$, MSWD $= 0.99$, $p = 0.52$) with an initial $^{87}Sr/^{86}Sr$ of $0.714 \pm 0.009$ (Supplementary Fig. C). All of the ages and initial ratios overlap with the published values (Kirkland et al., 2007) within 2σ uncertainty.

A small round robin analytical run consisting of ~20 standards preceded analytical runs to monitor long-term stability, and

overall data integrity. Data were reduced in Iolite (Paton et al., 2011) and in-house Excel macros. Analyses that crosscut multiple minerals or mineral generations at depth were excluded. Rb-Sr isochrons and ages were computed using Isoplot 4.15 (Ludwig, 2012), with the decay constant after Villa et al. (2015). All uncertainties presented in the text are presented at 95% confidence. Full isotopic data for the samples and reference materials are given in supplementary Table B.

**5.0  Results**

**5.1  Microstructure and mineral paragenesis from the Tropicana gold mine**

In the ten samples from the Tropicana gold mine, two mineral assemblages are identified and linked to two distinct microstructures (Figs. 2, 3). Mineral assemblage 1 comprises perthitic K-feldspar, plagioclase, quartz, euhedral biotite 1 (1st generation), apatite 1 (1st generation), zircon, monazite and Au-bearing pyrite 1 (1st generation). Both Au-bearing pyrite and

apatite 1 occur as inclusions in K-feldspar (Fig. 3a, e). Lamellae of rutile within coarse-grained biotite 1 were previously identified by Doyle et al. (2015) but were not observed in this study. The coarse-grained microstructure and associated mineral assemblage 1 is rarely preserved in the ore zone due to the low-temperature and high-strain shearing (Fig. 2a, b, c). However, main rock-forming minerals are preserved either in low-strain domains or as porphyroclasts within mylonites (Fig. 2a, b, c).

The fine-grained microstructure and associated mineral assemblage 2 is related to localized brittle to brittle-ductile strain, overprinting assemblage 1 (D3 of Blenkinsop and Doyle, 2014). The brittle strain affects perthite to form a so-called crackle breccia (Blenkinsop and Doyle, 2014). The brittle–ductile strain has reworked quartz and biotite to form transitional microstructures between jigsaw puzzle breccia and core-and-mantle microstructure (Fig. 2c), while plagioclase broke down to a sericite mesh (Fig. 2a, b, c). Dynamic recrystallization was accompanied by the ingress of hydrothermal fluids that

precipitated carbonates, pyrite and phengite (Fig. 2a, b, c) and breaks down perthite to albite along fractures (Fig. 2b). The low-strain microstructure represents the main target for our *in situ* dating of an early biotite 1 and dynamically recrystallized biotite 2 (Fig. 3a–d). In the high-strain zone, sericite forms interconnected matrix to porphyroclasts of perthite, quartz and biotite (Fig. 2d). A potential second generation of apatite (apatite 2) is also found interstitially together with assemblage 2 minerals (Fig. 3f).





## 5.2 *In situ* Rb-Sr geochronological data

### 5.2.1 Biotite

Biotite was analyzed from all samples from the Tropicana gold mine (Table 1). According to the mineral paragenesis, biotite was sub-divided into two texturally distinct sub-populations: (i) large, subhedral to euhedral grains associated with assemblage 1, and (ii) recrystallized, fine-grained crystals associated with assemblage 2 (Fig. 3).

The older component – texturally part of assemblage 1 – does not define a single population and has relatively low $^{87}$Rb/$^{86}$Sr (8–620, mean = 171) compared to assemblage 2 biotite (Fig. 4). Assuming initial $^{87}$Sr/$^{86}$Sr ratios of 0.7045–0.7058 (as defined by apatite, see below), the variable Rb-Sr ratios yield broadly linear trends with dates between ca. 2400 and 1500 Ma for the four samples (Fig. 4). However, significant variation is observed in all samples. The analyses with the oldest dates (on average) are from samples with limited presence of recrystallized biotite 2. Conversely, samples with younger dates have a more significant proportion of biotite 2.

Spot profiles across two large biotite 1 grains were performed to ascertain if there was systematic age variation from core to rim (Fig. 5). The transects show that there is no significant variation in model ages observed across the large grains, with the exception of some younger dates towards the very edges (e.g., analysis 19 in BSD114-541C or analyses 1–2 in TPD542-371-C) or where fractures were intersected with minor recrystallized biotite 2 (Fig. 5) .

Analyses from the second phase of biotite (biotite 2) yielded statistically valid isochrons in every sample and shows a wider range of $^{87}$Rb/$^{86}$Sr ratios (up to 950, mean = 210, Fig. 4). Ages computed from Rb/Sr isochrons are 1165 ± 140 Ma, 1227 ± 100 Ma, 1211 ± 19 Ma, to 1208 ± 36 Ma for the four samples from the Tropicana gold mine ($p > 0.05$ in all cases, Fig. 4). For three of the samples where a putative second generation of apatite may be coeval with biotite 2 (see section 5.1), it is possible to compute an isochron with both apatite and biotite 2. Combined apatite and biotite 2 yields isochrons of 1222 ± 37, 1241 ± 33 Ma and 1205 ± 15 Ma ($p > 0.05$ in all cases, Fig. 4) but with ages that overlap in uncertainty if apatite is not used in the calculation. There is no systematic variation in ages between samples or between Tropicana gold mine pits.

### 5.2.2 Phengite

Phengite was analyzed from three of the four samples from the Tropicana gold mine. Phengite, associated with assemblage 2 in the Tropicana gold mine, yielded single, linear Rb/Sr trends with low to moderate $^{87}$Rb/$^{86}$Sr values (0.2–32, mean = 13; Fig. 6). Sample TPD542-371-C yielded a statistically-reliable but imprecise age of 1212 ± 98 Ma ($n = 6$, MSWD = 1.3, $p = 0.28$, Fig. 6b). The two other samples, both from the Havana Pit, yielded broadly linear trends with age estimates of ca. 1220 and 1280 Ma but with overdispersion for a single population (MSWD = 3.5–3.7, $p < 0.05$; Fig. 6c, d). Notwithstanding the data scatter, on a given sample, the phengite age estimates are similar to those obtained via biotite Rb/Sr (cf. Fig. 4).



### 5.2.3 Muscovite

Muscovite was present as euhedral crystals in the New Zebra satellite deposit. Muscovite showed only minor spread in $^{87}Rb/^{86}Sr$ (2.0–3.4, **Fig. 6**a). Consequently, it yielded a statistically-valid but imprecise isochron of $1255 \pm 170$ Ma ($n = 37$,
MSWD = 1.14, $p = 0.26$) with an initial $^{87}Sr/^{86}Sr$ intercept of $0.7116 \pm 0.0062$ (**Fig. 6**a).

### 5.2.4 Apatite

As apatite has negligible Rb, there is no modification of initial $^{87}Sr/^{86}Sr$ from any radiogenic decay of $^{87}Rb$. Thus, the measured $^{87}Sr/^{86}Sr$ is equivalent to the initial $^{87}Sr/^{86}Sr$ ratio at the time of (re)crystallization. Apatite 1 was analyzed in three samples, yielding $^{87}Sr/^{86}Sr_{(i)}$ ratios between $0.7045 \pm 0.0012$ to $0.7058 \pm 0.0039$ ($p > 0.05$ in all samples; Fig. 7a, c, d). Apatite 2 was
analyzed in three samples, yielding $^{87}Sr/^{86}Sr_{(i)}$ ratios between $0.7053 \pm 0.0012$ to $0.7092 \pm 0.0033$ ($p > 0.05$ in all samples, Table 1, Fig. 7a, b, d). In the two samples where both assemblages of apatite were analyzed, BSD114-514-C yielded initial $^{87}Sr/^{86}Sr$ that overlapped within 2σ error but HDD254-711-C yielded more radiogenic values for apatite 2 compared to apatite 1 (Fig. 7a, d).

## 6.0  Discussion

## 6.1  Ages recorded by the Rb-Sr geochronometers in the Tropicana Zone

Two distinct temporal patterns are identified in the Rb-Sr results (Figs. 4–7). We examine below the two isotopic patterns to evaluate their significance and assess the formation mechanisms.

### 6.1.1 Assemblage 1: ca. 2530 Ma

The first generation of biotite grains consistently shows a scatter of data and a variation in mean isochron ages, with a strong
textural control on these age estimates. Samples with dominantly euhedral biotite grains and minimal recrystallized biotite 2 (e.g., BSD114–541-C) yielded a mean age estimate of ca. 2500 Ma (Fig. 4a). Conversely, where the majority of biotite 1 has been recrystallized to biotite 2 (e.g., HDD077-422-C), the age estimates from biotite 1 are as young as ca. 1500 Ma (Fig. 4c). The most likely explanation for this data scatter is variable loss of Rb and Sr (especially radiogenic $^{87}Sr$) during (partial) resetting of the Rb-Sr isotopic system (Matheney et al., 1990;Kalt et al., 1994;Evans et al., 1995;Eberlei et al., 2015). The
mechanism for this phenomenon is related to the crystallographic position and relative stability of $^{87}Rb$ and $^{86}Sr$ vs $^{87}Sr$. In trioctahedral micas (e.g., biotite and phlogopite), both $^{87}Rb$ (1.72 Å) and $^{86}Sr$ (1.44 Å) are situated within a large 12-fold coordinated X-site (Shannon, 1976), where Rb can readily exchange for K and Na, and Sr can exchange with Ca (Zussman, 1979). However, once $^{87}Rb$ has decayed to $^{87}Sr$ in a K or Na site, there is a significant decrease in ionic radius from 1.72 to 1.44 Å (16%; Shannon, 1976), which makes $^{87}Sr$ prone to mobility. Therefore, in fluid-mediated recrystallization during





hydrothermal or metamorphic events, $^{87}$Sr can escape more readily from the crystal lattice than $^{87}$Rb or $^{86}$Sr into the percolating
        fluid.

        Considering that the excess scatter in the first assemblage biotite grains is due to partial resetting linked to dynamic
        recrystallization, it follows that the oldest biotite grains can yield a minimum age for the first event. In the samples where the
crystals are freshest, coarsest and least overprinted by shearing and assemblage 2, the oldest biotite grains provide a minimum
        age of 2535 ± 18 Ma for assemblage 1 (Fig. 4a). This is consistent within error with biotite $^{40}$Ar/$^{39}$Ar (2531 ± 14 Ma) and
        pyrite Re-Os ages (2505 ± 50 Ma), and broadly compatible with the oldest W-rich rutile U-Pb dates of ca. 2539–2479 Ma
        obtained from similar samples in the Tropicana gold mine (Fig. 9; Doyle et al., 2015).

        **6.1.2 Assemblage 2: ca. 1210 Ma**

Ages of the recrystallized biotite grains from assemblage 2 are within uncertainty of one another for all samples from the
        Tropicana gold mine. Considering the relative proximity of all samples within the Tropicana gold mine, it is unlikely that the
        duration of a hydrothermal event associated with dynamic recrystallization of biotite would have exceeded the uncertainty of
        the Rb-Sr geochronometer. We therefore calculate a single weighted mean age of 1207 ± 12 Ma (*n* = 62, MSWD = 0.91, *p* =
        0.68) for all available analysis of this fabric (Fig. 8), and consider this age to record a synchronous event across the Tropicana
gold mine. Assuming biotite 2 grains were coeval with apatite 2, a better initial $^{87}$Sr/$^{86}$ intercept can be defined, relative to one
        generated from a free-regressed biotite 2 isochron. Hence, we have also computed a weighted mean age with apatite 2 and
        biotite 2 at 1212 ± 9 Ma (*n* = 102, MSWD = 1.3, *p* = 0.05), overlapping within uncertainty with the biotite-only regressed age
        (Fig. 8).

The phengite ages consistently yield similar ages within 2σ of the biotite 2 ages, albeit with excess scatter, implying that biotite
        2 and phengite are coeval (Figs. 6, 8). This is consistent with the mineral paragenetic sequence (Fig. 2). The overdispersion in
        the phengite data could be a consequence of heterogeneous minerals or differences in the matrix between the phengite and the
        phlogopite primary standard Mica-Mg (see section 4.2). Alternatively, the overdispersion may be a real geological
        phenomenon, implying that phengite formed over a protracted period.


        The only sample that contained euhedral muscovite, from the New Zebra deposit, yielded an age of 1255 ± 170 Ma that,
        although imprecise, also overlaps with the second phase of biotite (biotite 2) and the phengite ages (Figs. 4, 6). Thus, all
        samples point towards a single event at ca. 1210 Ma.

The ca. 1210 Ma event could either represent a distinct, fluid-flow episode that recrystallized biotite, formed phengite and
        yielded muscovite, or it may record an exhumation event that cooled the Rb-Sr geochronometers to below their closure
        temperatures. The implication with the latter is that the fluid flow event occurred prior to exhumation.



### 6.2 Implications for metamorphic, hydrothermal and mineralization events in the Albany–Fraser Orogen

Two ages at ca. 2530 Ma and 1210 Ma have been previously linked to events in the Albany–Fraser Orogen, although the latter had not been previously identified in the Tropicana Zone. The discovery of the Mesoproterozoic age at Tropicana needs to be explored in terms of implications for structural and hydrothermal evolution, and metallogenesis.

The ca. 2530 Ma age is only known from the Tropicana Zone of the Albany–Fraser Orogen (Doyle et al., 2015;Kirkland et al.,
2015). Previous workers have interpreted the ca. 2530 Ma age as a distinct hydrothermal event at greenschist facies conditions during D3 shearing, which was also associated with Au mineralization (Blenkinsop and Doyle, 2014;Doyle et al., 2015;Occhipinti et al., 2018). This ca. 2530 Ma event is postulated to have followed a protracted period of granulite-facies metamorphism from ca. 2640 to 2530 Ma as part of the Atlantis event (Kirkland et al., 2015;Doyle et al., 2015).

Here, we propose an alternative plausible scenario, namely that the ca. 2530 Ma age represents the timing of cooling below the closure temperatures of the various geochronometers, including U-Pb rutile ≈ ca. 2539–2479 Ma, Re-Os pyrite = 2505 ± 50 Ma, Rb-Sr biotite 1 ≥ 2535 ± 18 Ma and $^{40}$Ar/$^{39}$Ar biotite = 2531 ± 14 Ma (Doyle et al., 2015 and this study). The closure temperatures for these minerals are low to moderate, in decreasing order: (i) Pb diffusion in rutile = 550–650 °C (Ewing et al., 2015;Kooijman et al., 2010), (ii) $^{187}$Os in pyrite closure = ~500 °C (Brenan et al., 2000), (iii) $^{87}$Sr diffusion in biotite = 300–
400 °C (Del Moro et al., 1982), and (iv) $^{40}$Ar diffusion in biotite = 280–350 °C (Harrison et al., 1985). Given that all the geochronometers are all broadly within error of ca. 2530 Ma (this study and Doyle et al., 2015), exhumation at ca. 2530 Ma would have been relatively faster than the preceding ~120 m.y. of the Atlantis Event. Such prolonged slow cooling followed by a relatively faster period of cooling is also observed in the core of the Yilgarn Craton (Goscombe et al., 2019). The rate of cooling and exhumation in Tropicana Zone is difficult to ascertain as the uncertainty on the various geochronometers is
insufficiently precise to develop a cooling curve. If the ca. 2530 Ma age recorded in the Tropicana Zone represents a cooling age rather than a distinct tectonic event, an important implication is that D3 shearing occurred post-2530 Ma.

We propose that D3 shearing instead occurred at ca. 1210 Ma. There are several lines of evidence to support this interpretation:
(1) Mineral assemblage 2 associated with the fine-grained microstructure is stable from low- to high-strain zones across
the Tropicana gold mine and does not show any reactivation/secondary dynamic recrystallization or mineral re-equilibration to support subsequent reactivation of the shear zone (Blenkinsop and Doyle, 2014). If only a single shearing event is implicated, this would have to occur during the formation of mineral assemblage 2 (i.e., 1210 Ma).
(2) Rb-Sr profiles across coarse-grained biotite 1 show limited resetting at grain edges, linked to dynamic recrystallization (Fig. 5), and also consistent with a single shearing event (i.e., 1210 Ma).



(3) If D3 shearing occurred at ca. 2530 Ma and it was reset at ca. 1210 Ma, one would expect a range of Proterozoic ages from biotite 2, which is clearly not the case (Fig. 8). One could argue that previous geochronological studies in the Tropicana gold mine pointed towards a potential mineralizing event at ca. 2000–1800 Ma on the basis of disturbed $^{40}$Ar/$^{39}$Ar spectra and Pb-Pb dates (Doyle et al., 2015). Additionally, quartz vein-related Au mineralization is implicated at ca. 2100 Ma for the Hercules and Atlantic gold prospects in the Tropicana Zone based on Re-Os pyrite

model ages (Kirkland et al., 2015). However, it is equally likely that the Pb-Pb and Re-Os pyrite dates (Doyle et al., 2015) represent mixed assemblage 1 and 2 populations, and do not represent distinct events. Disturbed $^{40}$Ar/$^{39}$Ar spectra are also notoriously unreliable (Baksi, 2007).

    (4) Finally, there is no unequivocal geological evidence for a distinct shearing event at ca. 2530 Ma (Doyle et al., 2015).

Our support for D3 shearing at ca. 1210 Ma does not necessarily mean that assemblage 2 was linked to the primary Au mineralization in the Tropicana deposit (Fig. 9). Mesoproterozoic orogens are typically poor in orogenic Au deposits across the world (e.g., Goldfarb et al., 2001), and the Albany–Fraser Orogen is no exception. Since the discovery of Tropicana in 2005 (Doyle et al., 2007;Kendall et al., 2007), it remains the only deposit with economic Au mineralization in the Albany-Fraser Orogen. Although Tropicana is unlike typical Archean lode gold deposits in the Yilgarn Craton (e.g., gold not directly

associated with quartz and carbonate veining; Kent et al., 1996;Cassidy et al., 1998), the Tropicana Zone experienced long-lived granulite facies metamorphism from ca. 2640 to 2530 Ma (Atlantis Event; Doyle et al., 2015;Kirkland et al., 2015). Such long-lived metamorphism might have efficiently reworked all textural indicators of primary mineralization. Indeed, a detailed microscale study of the gold compartment in the Tropicana gold mine has demonstrated gold and telluride inclusions within granulite facies coarse-grained material from assemblage 1 (Hardwick, 2020). Following these observations, it seems that D3

shearing and alteration minerals of assemblage 2 masks the controls on primary gold mineralization.

There are certainly Proterozoic occurrences of sub-economic Au mineralization in the rest of the Tropicana Zone and possibly the wider Albany–Fraser Orogen (Figs. 1b, 6a, 9). Gold prospects such as New Zebra, Iceberg or Angel Eyes lack obvious Archean events (Fig. 6a) but still show elevated Au. Therefore, it is probable that subordinate, secondary Au mineralization in

the Tropicana Zone is associated with the D3 shearing event and mineral assemblage 2 formation at ca. 1210 Ma (Fig. 9), but it is uncertain whether the Au is remobilized from an Archean source or was only introduced into the Mesoproterozoic crust at ca. 1210 Ma.

The ca. 1210 Ma event in the Albany–Fraser Orogen is known from both the early stages of the Albany–Fraser Orogeny Stage

II (Clark et al., 2000;Kirkland et al., 2011;Spaggiari et al., 2014) and the widespread intrusions of the Marnda Moorn dyke swarm (Wang et al., 2014;Wingate and Pidgeon, 2005;Dawson et al., 2003). We favour the association of the D3 shearing with Stage II of the Albany–Fraser Orogeny given the macro- and microstructural characteristics of the Tropicana gold mine.



## 7.0 Conclusions

*In situ* Rb-Sr geochronology from two assemblages of biotite in the Tropicana gold mine yielded ages of 2535 ± 18 Ma and
1212 ± 9 Ma. The former overlaps with $^{40}$Ar/$^{39}$Ar biotite, Re-Os pyrite and U-Pb rutile ages obtained in a previous study,
whilst the latter is the first record of a Mesoproterozoic age in the Tropicana Zone. We propose that the ca 2530 Ma represents
cooling of the Yilgarn Craton after granulite facies metamorphism and that the ca. 1210 Ma represent a distinct shearing event
(D3 of Blenkinsop and Doyle, 2014), potentially associated with Au mineralization. Considering the likely association of the
ca. 1210 Ma event with major shearing, we consider the most likely cause of this Mesoproterozoic event to be Stage II of the
Albany–Fraser Orogeny. At present, the *in situ* Rb-Sr method is the only technique that could have revealed these two age
populations without foregoing textural context.

## 8.0 Acknowledgements

AngloGold Ashanti Australia is thanked for funding this project. The Tescan Mira3 TIMA with four PulsTor SDD X-ray
detectors was acquired through Australian Research Council LIEF grant LE140100150. This study was enabled by AuScope
and the Australian Government via the National Collaborative Research Infrastructure Strategy (NCRIS). The 8900 triple
quadrupole was obtained via funding from the Curtin University Research Office. J. Savage and B. Hardwick are
acknowledged for constructive discussions about the Tropicana deposit.

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



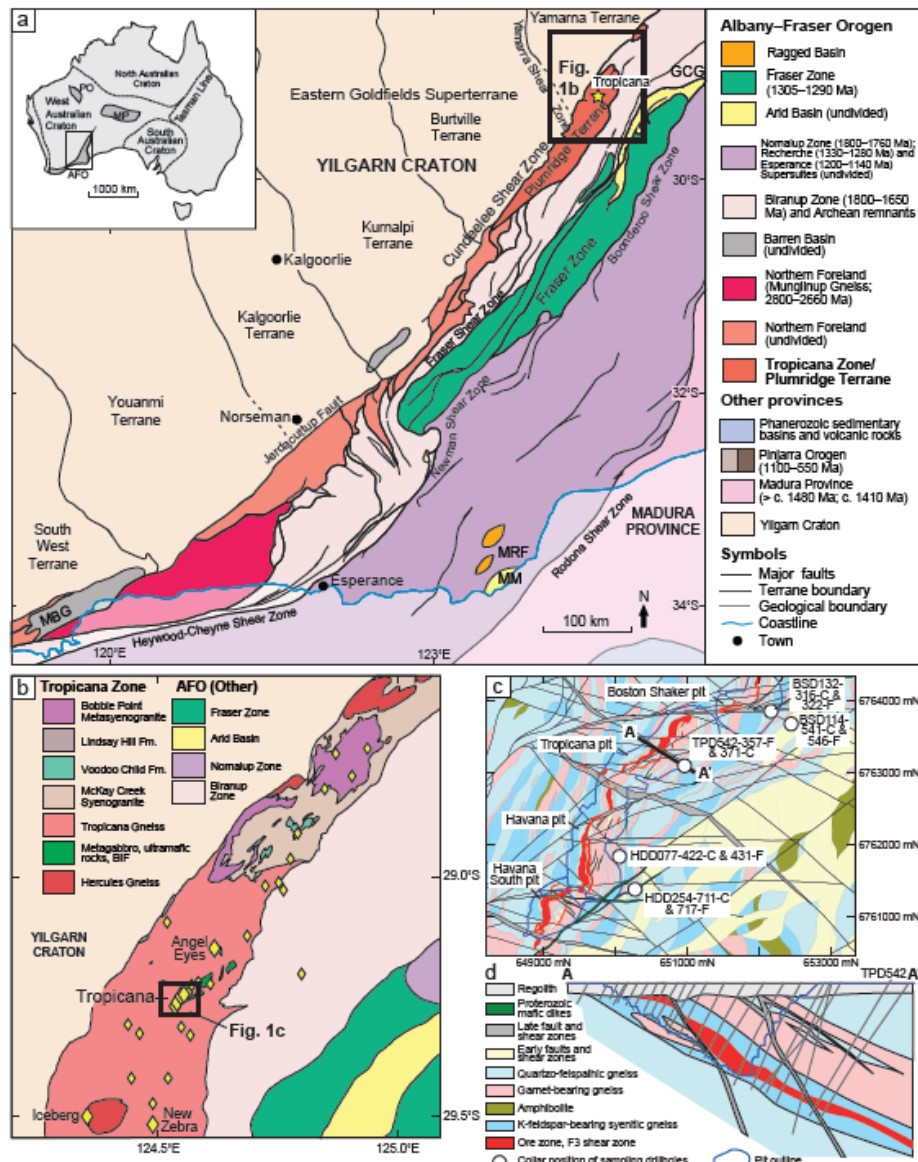

**Fig. 1: (a) Simplified, pre-Mesozoic interpreted bedrock geology of the Albany–Fraser Orogen, modified from Spaggiari et al. (2015),**

**Doyle et al. (2015) and Scibiorski et al. (2016). Abbreviations: GCG—Gwynne Creek Gneiss; MBG—Mount Barren Group; MM—
Malcolm Metamorphics; MRF—Mount Ragged Formation; WF—Woodline Formation. (b) Interpreted basement map of the
Tropicana Zone/Plumridge Terrane, modified from Kirkland et al. (2015). Gold deposits and prospects are shown from the
Geological Survey of Western Australia MINEDEX database, with studied locations labelled. (c) Interpreted basement map of the
Tropicana gold mine from internal AngloGold Ashanti maps, showing locations of drill holes analyzed in this study. Note the**

**Tropicana and Havana pits have now joined. (d) Interpreted cross-section across the ore-bearing and F3 shear zone of the Tropicana
gold mine from AngloGold Ashanti, showing locations of drill holes (pale grey lines). All maps use GDA 1994 geodetic datum with
(c) using MGA zone 51 projection.**







**Fig. 2: Dual transmitted and reflected light, cross-polarized photomicrographs showing characteristic low-strain (a–c) and high-**
**strain (d) microstructures and mineral relationships of studied samples. (a) Sample BSD132-316-C shows fractured perthitic**
**feldspar from Assemblage 1 in the centre with carbonate, pyrite and quartz from Assemblage 2 filling the fracture. Quartz and**
**biotite 1 show formation of sub-grains (Biotite 2). (b) Sample HDD254A-711-C, a carbonate from Assemblage 2 fills shear fractures**
**that crosscut perthitic feldspar from Assemblage 1. Reaction rims along fractures show breakdown of perthite to albite. (c) Sample**
**BSD132-316-C shows brittle–ductile deformation of large quartz and biotite 1 (bottom right) grains and precipitation of pyrite in**
**strain shadows. (d) Sample HDD077-431-F shows porphyroclasts of perthitic feldspar within a mylonitic matrix consisting of biotite**
**2, quartz, carbonate, pyrite and sericite.**



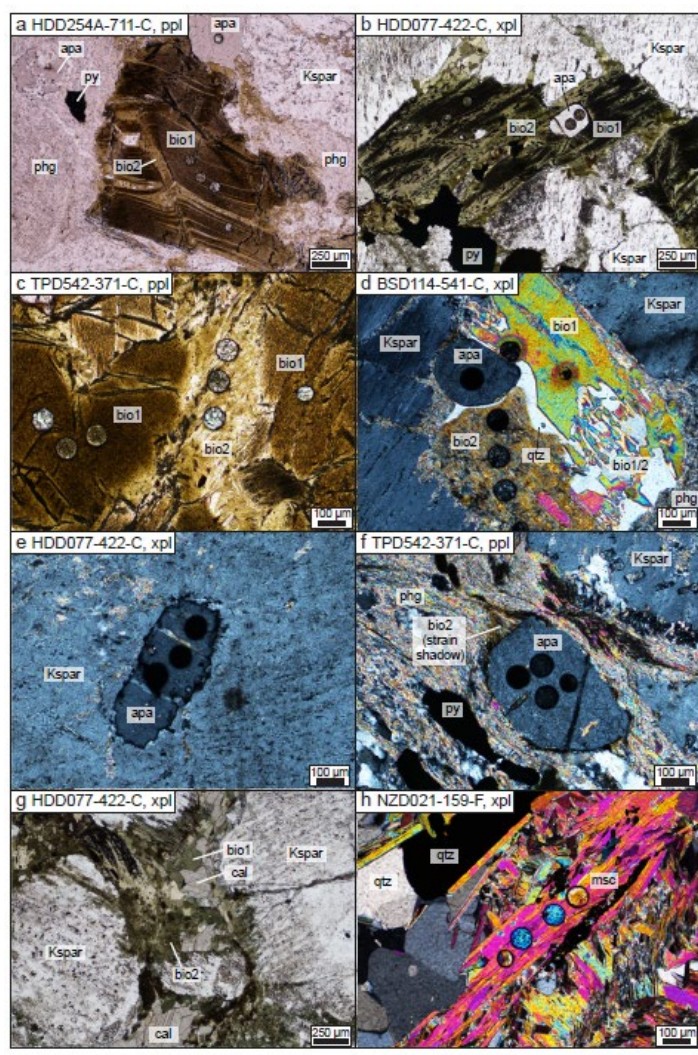

**Fig. 3: Thin section photomicrographs in transmitted light showing detailed mineral. (a) Biotite 2 recrystallized sub-parallel to axial plane of folded biotite 1, situated in association with phengite and interstitial to K-feldspar phenocrysts, from HDD254A-711-C. (b) Biotite 1 variably recrystallized to biotite 2, and associated with pyrite and apatite, all interstitial to K-feldspar phenocrysts from HDD077-422-C. (c) Recrystallized biotite 2 corridor between biotite 1 in TPD542-371-C. (d) Biotite in various stages of recrystallization, from euhedral biotite 1 to partly recrystallized biotite 1/2 to fully recrystallized biotite 2, from BSD114-541-C. Note association of unstrained quartz with recrystallization. (e) Apatite (magmatic?; 1?) fully enclosed in K-feldspar phenocryst from HDD077-422-C. (f) Apatite (2) porphyroclast with rare strain shadow of biotite 2 and phengite, implying locally biotite 2 > apatite > phengite, from TPD542-371-C. Also note close association of pyrite. (g) Carbonate ingress with recrystallization of biotite 1 to biotite, interstitial to K-feldspar from HDD077-422-C. (h) euhedral muscovite in association with quartz, from NZD021-159-F. apa = apatite, bio1 = euhedral, first generation biotite, bio1/2 = partly recrystallized biotite 1, bio2 = recrystallized biotite 1, cal = calcite, Kspar = K-feldspar, msc = muscovite, phg = phengite, py = pyrite, qtz = quartz, ppl = plane-polarized, xpl = cross-polarized. Circular holes are laser ablation pits of 87 or 60 μm diameters.**



**Fig. 4: Biotite Rb–Sr isochrons for individual samples, with red data corresponding to assemblage 1 and green data corresponding to assemblage 2 biotite. Grey data is considered to be part of assemblage 1 but with loss of radiogenic Sr. All error ellipses are plotted at 2σ. The 2530 Ma model age is based off Fig. 4a, assuming an initial $^{87}Sr/^{86}Sr$ of 0.7045. Note that for assemblage 2, ages are calculated both solely with biotite 2 and with biotite 2 and apatite combined (see section 3 and discussion).**





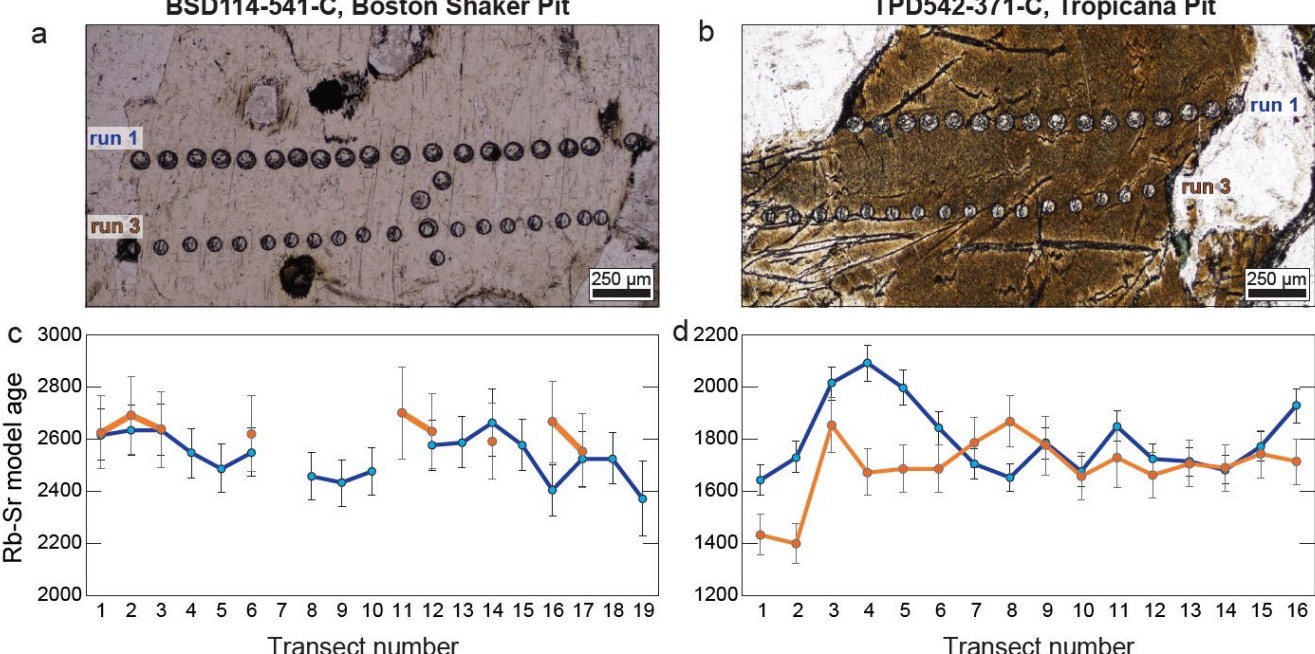

Fig. 5: Transects across two large, euhedral biotite grains. (a, b) Transmitted, plane-polarized photomicrographs of part of the euhedral grains, showing 87 µm (run 1) and 64 µm (run 3) laser ablation pits along the grains. (c, d) Rb-Sr model ages for each spot in the transect across two runs. Model ages were computed using $^{87}Sr/^{86}Sr$ of apatite 1 in corresponding sample (see Fig. 7).



Fig. 6: Phengite and muscovite Rb-Sr isochrons for individual samples, with green data corresponding to assemblage 2 phengite or muscovite. All error ellipses are plotted at 2σ. Ages in italics are estimates only due to overdispersion for a single population.







Fig. 7: Apatite weighted mean $^{87}Sr/^{86}Sr$. Due to the lack of Rb in apatite, measured $^{87}Sr/^{86}Sr$ values can be considered as initial ratios.



**Fig. 8: Combined isochron of biotite 2 analyses from all four samples in the Tropicana gold mine, computed with and without apatite 2.**





**Fig. 9: Synthetic time–space plot showing geological history of the Tropicana Zone, modified from Spaggiari et al. (2015) and**

**Kirkland et al. (2015). The timing of mineralization in the Tropicana Zone is uncertain (see discussion).**



| Deposit/pit | Easting (m) | Northing (m) | RL (m) | Sample ID | Assem-blage | Main minerals | Rb-Sr phases | Dated sessions |
|---|---|---|---|---|---|---|---|---|
| Boston Shaker | 6763834 | 652321 | 354 | BSD 132-316-C | 1 | Coarse-grained quartz, K-feldspar; medium-grained pyrite; fine-grained apatite, monazite; interstitial biotite, (Fe-rich) calcite, phengite | Biotite 1/2, Phengite | - |
| Boston Shaker | 6763834 | 652321 | 354 | BSD 132-322-F | 2 | Medium-grained perthitic K-feldspar, albite, pyrite, quartz; quartz veining; interstitial (Fe-rich) calcite, phengite and biotite | Biotite 1/2, Phengite | - |
| Boston Shaker | 6763644 | 652657 | 348 | BSD 114-541-C | 1 | Coarse-grained K-feldspar; medium-grained plagioclase; apatite; interstitial biotite, quartz, actinolite; fine-grained pyrite | Biotite 1/2, Phengite, Apatite 1/2 | 1,3 |
| Boston Shaker | 6763644 | 652657 | 348 | BSD 114-546-F | 2 | Medium-grained plagioclase, quartz; fine-grained pyrite, ankerite, foliation-parallel phengite | Biotite 1/2, Phengite | - |
| Tropicana | 6763087 | 651144 | 344 | TPD 542-357-F | 2 | Strongly foliated, defined by fine-grained plagioclase, pyrite, biotite and phengite; rare quartz, calcite and apatite | Biotite 1/2, Phengite | - |
| Tropicana | 6763087 | 651144 | 344 | TPD 542-371-C | 1 | Coarse-grained K-feldspar, medium-grained biotite, apatite, plagioclase, quartz; fine-grained pyrite; interstitial biotite, phengite | Biotite 1/2, Phengite, Apatite 2 | **1,3** |
| Havana | 6761694 | 650183 | 362 | HDD 077-422-C | 1 | Coarse-grained (perthitic) K-feldspar; medium-grained plagioclase; fine-grained apatite, pyrite, zircon; interstitial biotite, phengite, quartz | Biotite 1/2, Phengite, Apatite 1 | 1,3 |
| Havana | 6761127 | 650536 | 362 | HDD 254-711-C | 1 | Coarse-grained (perthitic) K-feldspar; medium-grained plagioclase, apatite, quartz, pyrite; interstitial biotite, phengite, dolomite | Biotite 2, Apatite 1/2 | 1,2,3 |
| Havana | 6761127 | 650536 | 362 | HDD 254-717-F | 2 | Medium-grained K-feldspar, quartz; fine-grained plagioclase, pyrite, apatite, dolomite; interstitial biotite, phengite | Biotite 1/2, Phengite, Apatite 2 | - |
| New Zebra | 6730709 | 642098 | 382 | NZD 021-159-F | 2 | Parasitically folded defined by quartz and muscovite; ankerite and calcite abundant; minor fine-grained rutile, pyrite, albite, apatite | Muscovite | 1,3 |
| Iceberg | 6733203 | 634266 | 385 | IBD 004-76-F | 2 | Strongly foliated, foliation defined by quartz, altered plagioclase, phengite, fine-grained pyrite | Phengite | - |
| Angel Eyes | 6772309 | 657438 | 316 | AERC 012D-156-F | 2 | Strongly foliated, two zones. Upper zone: medium-grained quartz; interstitial phengite, ankerite, pyrite. Lower zone: very fine-grained quartz, phengite, pyrite and ankerite, apatite | Phengite, Apatite 2 | - |

**Table 1: List of samples dated in this study. TGM = Tropicana gold mine. C = Coarse-grained. F = Fine-grained. Rb/Sr phases indicates the minerals that were dated in each sample, with 1 and 2 corresponding to assemblage 1 and 2, respectively. Eastings, Northings and Reduced Level (RL = Z) are in GDA 1994, MGA zone 51 projection for the drill hole collar.**



**Supplementary Table A: Rb-Sr geochronology parameters during the three analytical sessions.**

**Supplementary Table B: Compendium of Rb-Sr data, including unknowns and reference materials.**

**Supplementary Fig. A: Graphic illustration of how a reaction cell sandwiched between two quadrupoles separates Rb from Sr**
**isotopes, as well as deals with interfering ions.**

**Supplementary Fig. B: Compilation of (i) core photos, (ii) transmitted, plane-polarized light image of full thin section, (iii) transmitted, cross-polarized light image of full thin section, and (iv) automated mineral analysis image of full thin sections, accompanied by legend. Yellow line in core photos indicates approximate position of thin section billet. All thin sections are 46 × 27**
**mm.**

**Supplementary Fig. C: Rb-Sr isochron showing identical ages within uncertainty between sessions 1, 2 and 3 for biotite 2 from HDD254-711-C (Tropicana, WA) and biotite from CK001B (Finnmark, Norway). Spots for these analyses were collected adjacent to each other (see Fig. 2 for examples). Ages for CK001B overlap with ages obtained by Rb-Sr solution ages (Kirkland et al., 2007).**