# Peer review of "Resolving multiple geological events using in situ Rb-Sr geochronology: implications for metallogenesis at Tropicana, Western Australia"

_Geochronology, 2020_

## Referee Comment (RC1) · Thomas Blenkinsop (Referee) · 3 Jun 2020

The aim of this paper is to demonstrate that in situ Rb/Sr dating has now become sophisticated enough to unravel different geological events. The study uses samples from the Tropicana area, which by now is quite well constrained in terms of regional and deposit scale geochronology. The paper is very clearly written and illustrated, and communicates its message very well. There is no doubt that it shows the power of the method, which is an exciting advance in geochronology and this aspect needs to be published.

[Figure]

The interpretation of the data with respect to tectonics and mineralization is more controversial. The older ages agree with previous geochronology (a strength of the paper), but they are interpreted in a new and different way as simple cooling ages rather than relating to the D3 deformation event. This is regarded as having the younger age (1210 Ma). The major reason for this is the interpretation that a single shearing event is seen in the microstructures, unlike the reactivation scenario previously postulated. This is not consistent with the change in kinematics of shear zones from D3 to D4, D5 that is documented in Blenkinsop and Doyle (2014). In that study, D3 shear zones were identified as having only biotite as the phyllosilicate phase, whereas most of the samples in this study have some muscovite/phengite, so they would be classified as D4 or D5 according to the previous work. It could therefore be suggested that none of the samples adequately dated a true D3 shear zone.

This study has the advantage of the TIMA images which may have revealed additional aspects of shear zones not seen in the 2014 study, so it may be that the petrographic distinction claimed previously is not real. However, there is a clear morphological difference between the shear zones with biotite and pyrite and those with phengite - the latter are generally wider, with much stronger fabrics. This can be seen for example in the differences between Fig. 7, a and b compared to c and d in Blenkinsop and Doyle (2014) There is clear structural evidence for shear zones that cross cut earlier biotite fabrics in the drill core, and there are clearly sets of shear zones with different kinematics (Blenkinsop and Doyle Fig. 14). It would be very strange if this was not the case in such a polymetamorphic setting, although that is not a strong argument. So it is a bold claim that there is no evidence for reactivation and that all deformation textures belong to a single event.

To substantiate the new interpretation, it would be useful to see some more microstructural analysis with kinematics and some more detailed photomicrographs of the dated samples.

In the end this debate is much less important than the geochronological aspects of the

paper, which seem really solid. The tectonic interpretation could therefore be presented with a more nuanced discussion, acknowledging the points above. It would be good to see this paper published, after dealing with this point. Tom Blenkinsop
* * *

---

## Referee Comment (RC2) · Thomas Zack (Referee) · 11 Jul 2020

General comments

This manuscript is a very good illustration of the new opportunities of in-situ Rb-Sr dating. It combines a range of state-of-the-art techniques relevant to mineral exploration (e.g., automated full thin section mineral identification) with texturally controlled in-situ Rb-Sr dating of micas. In the following a range of suggestions are presented how to improve this contribution. As it is a fully public review (something I need to get used

to), I hope I can also convince other colleagues to follow some of those suggestions.

1.) Novelty of contribution: As it is stated in the introduction, it is not the first publication dating two generations of mineral assemblages in the same sample. Therefore please remove "for the first time" in line 16 in the abstract. Instead, it could be stressed stronger that this study demonstrates that in-situ Rb-Sr dating allows tying microtextures to several geological events, as correctly stated in the last line of the abstract.

2.) Better microtextural documentation: although a big strength of this manuscript is the combination of microtexture and dating, this connection could and should easily be improved by adding BSE images of selected areas where LA-ICP-MS spots are visible. In general, BSE images give a clearer overview of where exactly spots were drilled and if any fractures, veinlets, inclusions were accidentally hit. In this study specifically, it is important for the reader to judge independently if a biotite 1 or 2 domain was sampled. A critical question is: are those mixed ages (in between 2.5 and 1.2 Ga) due to a partly resetting or by a mixed analysis? Figure 3c is a good example. The biotite 2 looks "bleached"- well, it could be different chemistry (less Fe and/or Ti), finer grain size or even a mixture of different minerals. This can only be solved by showing higher magnification images by BSE (alternatively reflected light can suffix; however, if there is a change in chemistry, this may be visible by a change in gray scale).

3.) Combination of ages and chemistry: do you have any additional information on the chemistry of the dated micas? Again, there could (should?) be a difference in chemistry if there are different mica generations (here perhaps visible in different Fe and/or Ti contents). This is not only relevant for distinguishing biotite 1 and 2, but also I do wonder why you distinguish between muscovite (s.s.) and phengite? Is it based on the Si-content? In this case microprobe data must exist, and it would be important to report. If not, then how? Unfortunately no other major (except Ca) and trace elements were measured along with the Rb and Sr isotopes by LA-ICP-MS. This is a pity as it is one of the unique advantages of quadrupole ICP-MS to combine dating and concentration determination (for the future. . .).

4.) How many age domains? Are there really "only" two significant ages extractable from the biotite data? A close inspection of figure 4b and 4d seems to show a clear linear trend of your "biotite 1" that falls on an isochron somewhere between the 2.5 Ga and 1.2 Ga. It could well be in this 1.8-1.6 Ga interval mentioned in chapter 2.1. I strongly recommend replotting the data from figure 4 in a histogram where x-axis is single spot model ages (like kernel density plots typical for detrital zircons). With a bit of statistics it can be tested if this third age is significant. For me it looks to regular to be a product of partial resetting (in contrast, the scatter visible by the grey circles in Figure 4a is a convincing example of partial resetting). IF such a third age can be established, the next question is: what distinguishes "biotite1" from samples in fig 4b and d from biotite from samples in fig 4a?

5.) Closure vs formation ages: as far as I can see, the difference between closure and formation ages has been handled in a succinct matter. Still, I think the manuscript will benefit by giving this topic a bit more prominence (e.g., defining those topics in the introduction and devoting a chapter in the discussion). It is currently a hot topic ("petrochronology" vs "thermochronology"), not always trivial to say which process is dominating in specific cases. Here you have (at least) two age populations of biotite not governed by volume diffusion (still, it is worth mentioning grain size effects on closure temperatures). Furthermore, muscovite is supposed to have higher closure temperatures than biotite, yet muscovite is younger than biotite 1, a clear indication that muscovite ages are formation ages.

Specific edits:

- abstract, line 16: instead of "K- and Rb-bearing", say "K-rich" or "Rb-bearing".

- abstract, line 20: replace "second assemblage" with "younger assemblage" (second and first are not defined in abstract)

- introduction, line 47: papers by Wolfgang Muller are good examples for texturally-controlled micromilling Rb-Sr dating (e.g., EPSL 180, 385-397).

- methods, line 193: instead "zero counts" (which do not exist), rather say "< xx cps".

- methods, line 217: please say a few more words on the biotite secondary standard CK001B. What chemistry does it have (annite vs phlogopite, etc)? Micas are a very large group with very different chemistry- it may turn out that "matrix" effects may even operate within a mica group, so it is good to have the chemistry specified.

- results, line 237-243: out of curiosity: do you have any constraints on the temperature conditions during mylonite formation? Would be nice to state if you can, as it will give important constraints that biotite 1 can statically survive a certain heating episode without diffusional resetting.

- results, line 271: "fractures were intercepted". Again, this is not clearly visible in the petrographic image. A BSE or reflected light image would be better.

- discussion, line 313-316: please delete the last two sentences, as they are too speculative!! You can ask Steve Reddy how messy cite occupancy of radiogenic isotopes are on an atomic level. Unless somebody is doing AFM on a mica, we simply do not know.

- figure 1b: please note that yellow diamonds are gold occurrences.

Best wishes, Thomas Zack
* * *

---

## Author Response (AR1)

**Authors' response to reviewers' comments**

**Manuscript title**: Resolving multiple geological events using in situ Rb-Sr geochronology: implications for metallogenesis at Tropicana, Western Australia

**Authors**: Hugo K. H. Olierook, Kai Rankenburg, Stanislav Ulrich, Christopher L. Kirkland, Noreen J. Evans, Stephen Brown, Brent I. A. McInnes, Alexander Prent, Jack Gillespie, Bradley McDonald, Miles Darragh

**Manuscript ID**: GCHRON-2020-7

**Date**: 10 August 2020

Dear editor,

Please find enclosed comments to the referees comments and your later comments on mineral chemistry. We have brought all key parts in line with the reviewer's constructive comments, particularly the petrographic context as highlighted by both reviewers.

We hope you will now find our manuscript satisfactory for publication. Please do not hesitate to contact us should any further clarification be required. We respond in detail to all reviewers' comments below.

Yours sincerely,

The authors

**Editor's comments:**

Dear Dr Olierook

I am generally satisfied on how you are planning to address the reviewers' comments. There are however two points that require additional attention and data.

1. The lack of mineral chemistry of the material analysed is not fully addressed. Mineral chemistry can be retrieved also after LA analysis. BSE images are a good start, but mineral chemistry is eventually needed. The same for phengite versus white mica, which can be analyzed after the fact; the petrographic analysis based on grain is not sufficiently robust. I thus request addition of analyses of mica domains beside the LA pits and of white mica versus phengite domains.

AGREE. We have used the mineral chemistry from the automated mineral analyses (TIMA), which is standardized to a Mn standard, to provide semi-quantitative mineral chemistry. These are of course not as robust as getting EPMA (or laser ablation) chemistry, but this at least provides a sense of the type of biotite, muscovite and phengite that was analyzed. We have added representative chemistry measurements in a supplementary table (B), added a new figure (Fig. 5) and added text in the section 5.1 of the results.

2. Composition of standard CK001B. I appreciate that details for this material are in review, but this does not prevent a statement to be added here of the general chemistry of the mica. We do not need the full analyses, which I suppose are included in the other publication, but at least an end members %.

AGREE. In the methods section, we have added "CK009 was collected < 50 km from CK001B, had similar biotite chemistry to CK001 and experienced equivalent Caledonian metamorphism (Kirkland et al., 2007). Biotite in sample CK009 is classified as magnesian siderophyllite according to an mgli-feal diagram (see Fig. 6 in Kirkland et al., 2007)"

Regards
Daniela Rubatto
Associate Editor

**Reviewers' comments:**

**Reviewer #1: Thomas Blenkinsop**

The aim of this paper is to demonstrate that in situ Rb/Sr dating has now become sophisticated enough to unravel different geological events. The study uses samples from the Tropicana area, which by now is quite well constrained in terms of regional and deposit scale geochronology. The paper is very clearly written and illustrated, and communicates its message very well. There is no doubt that it shows the power of the method, which is an exciting advance in geochronology and this aspect needs to be published.

We thank the reviewer for their time in reviewing this contribution.

The interpretation of the data with respect to tectonics and mineralisation is more controversial. The older ages agree with previous geochronology (a strength of the paper), but they are interpreted in a new and different way as simple cooling ages rather than relating to the D3 deformation event. This is regarded as having the younger age (1210 Ma). The major reason for this is the interpretation that a single shearing event is seen in the microstructures, unlike the reactivation scenario previously postulated. This is not consistent with the change in kinematics of shear zones from D3 to D4, D5 that is documented in Blenkinsop and Doyle (2014). In that study, D3 shear zones were identified as having only biotite as the phyllosilicate phase, whereas most of the sam- ples in this study have some muscovite/phengite, so they would be classified as D4 or D5 according to the previous work. It could therefore be suggested that none of the samples adequately dated a true D3 shear zone.

PARTLY AGREE. We agree with the disputed aspect of reactivation. This work is not focused on detailed microstructural and petrological study, that is required to assess reactivation scenario. Therefore, it is possible that the reactivation of the greenschist facies D3/D4 shears took place during Mesoproterozoic. We have corrected it in the text, accordingly citing Blenkinsop and Doyle (2014). However, we firmly believe that a low-strain brittle-ductile microstructure we used for dating both Biotite 1 and 2 represents the D3 event as described by Blenkinsop and Doyle (2014). A more discussion to this point is in the following point.

This study has the advantage of the TIMA images which may have revealed additional aspects of shear zones not seen in the 2014 study, so it may be that the petrographic distinction claimed previously is not real. However, there is a clear morphological difference between the shear zones with biotite and pyrite and those with phengite - the latter are generally wider, with much stronger fabrics. This can be seen for example in the differences between Fig. 7, a and b compared to c and d in Blenkinsop and Doyle (2014) There is clear structural evidence for shear zones that cross cut earlier biotite fabrics in the drill core, and there are clearly sets of shear zones with different kinematics (Blenkinsop and Doyle Fig. 14). It would be very strange if this was not the case in such a polymetamorphic setting, although that is not a strong argument. So it is a bold claim that there is no evidence for reactivation and that all deformation textures belong to a single event. To substantiate the new interpretation, it would be useful to see some more microstructural analysis with kinematics and some more detailed photomicrographs of the dated samples.

DISAGREE. We went through the paper of Blenkinsop and Doyle (2014) carefully again. Unfortunately, none of the figures shows a clear cross-cutting relationship between D3 and

D4 textures. Also, it is quite intriguing that both events show the same kinematics of NE-SW shortening in their paper (Table 1 for D3 event and see text in Page 198 for the D4 event). It is necessary to emphasise that the work of Blenkinsop and Doyle (2014) is based on a structural analysis of drill holes when pits were not open. Consequently, it is likely that a spatial and temporal relationship between D3 extensional textures and D4 shears could not have been adequately assessed.

We see a possibility that a variable amount of muscovite/phengite might be a function of variable plagioclase content of the host syenitic gneiss. An easy breakdown of plagioclase to micas localises strain within anastomosing and simple shear-dominated D4 ductile shears, while K-feldspar dominated domains show a more brittle response within an apparent low-strain and pure shear-dominated domain. A presence of carbonate and euhedral pyrite in both microstructures support the coincident development of the D3 and D4 microstructures. If only a single shearing event is implicated, this would have to occur during the formation of mineral assemblage 2 (i.e., 1210 Ma). Although the question of a possible reactivation is not the primary goal of this paper, it is likely that some reactivation of D3-D4 fabric might occur during localised D5 event as suggested by Blenkinsop and Doyle (2014). A presence of carbonate and euhedral pyrite in both microstructures support the coincident development of the D3 and D4 microstructures.

In the end this debate is much less important than the geochronological aspects of the paper, which seem really solid. The tectonic interpretation could therefore be presented with a more nuanced discussion, acknowledging the points above. It would be good to see this paper published, after dealing with this point. Tom Blenkinsop

AGREE. We thank the reviewer again for his constructive comments. We appreciate his knowledgeable insights to the geology of Tropicana deposit very much during this review as well as at the time when he worked on kinematics from drill holes. We realise how difficult it is to resolve the structural story in a remote and very poorly known area without outcrops and open pit observations. We have corrected our discussion to reflect this appreciation.

**Reviewer #2: Thomas Zack**

This manuscript is a very good illustration of the new opportunities of in-situ Rb-Sr dat- ing. It combines a range of state-of-the-art techniques relevant to mineral exploration (e.g., automated full thin section mineral identification) with texturally controlled in-situ Rb-Sr dating of micas. In the following a range of suggestions are presented how to improve this contribution. As it is a fully public review (something I need to get used to), I hope I can also convince other colleagues to follow some of those suggestions.

We thank the reviewer for their time to provide insightful comments for this paper.

1.) Novelty of contribution: As it is stated in the introduction, it is not the first publi- cation dating two generations of mineral assemblages in the same sample. Therefore please remove "for the first time" in line 16 in the abstract. Instead, it could be stressed stronger that this study demonstrates that in-situ Rb-Sr dating allows tying microtex- tures to several geological events, as correctly stated in the last line of the abstract.

AGREE. This has been removed.

2.) Better microtextural documentation: although a big strength of this manuscript is the combination of microtexture and dating, this connection could and should easily be improved by adding BSE images of selected areas where LA-ICP-MS spots are visible. In general, BSE images give a clearer overview of where exactly spots were drilled and if any fractures, veinlets, inclusions were accidentally hit. In this study specifically, it is important for the reader to judge independently if a biotite 1 or 2 domain was sampled. A critical question is: are those mixed ages (in between 2.5 and 1.2 Ga) due to a partly resetting or by a mixed analysis? Figure 3c is a good example. The biotite 2 looks "bleached"- well, it could be different chemistry (less Fe and/or Ti), finer grain size or even a mixture of different minerals. This can only be solved by showing higher magnification images by BSE (alternatively reflected light can suffix; however, if there is a change in chemistry, this may be visible by a change in gray scale).

AGREE. We thank the reviewer for raising this point – it has yielded some interesting discoveries. BSE images have been added as a new figure (4) and across the transects. Biotite 1 in the ca. 2.5 Ga samples is really 'clean', whereas biotite 1 in the other three samples with younger apparent isochron dates show exsolution lamellae of rutile that probably occurred during the formation of biotite 2. Thus, these laser spots have hit physical mixtures of biotite 1 and 2, with the ages younger than 2.5 Ga having no geological significance.

3.) Combination of ages and chemistry: do you have any additional information on the chemistry of the dated micas? Again, there could (should?) be a difference in chemistry if there are different mica generations (here perhaps visible in different Fe and/or Ti contents). This is not only relevant for distinguishing biotite 1 and 2, but also I do wonder why you distinguish between muscovite (s.s.) and phengite? Is it based on the Si-content? In this case microprobe data must exist, and it would be important to report. If not, then how? Unfortunately no other major (except Ca) and trace elements were measured along with the Rb and Sr isotopes by LA-ICP-MS. This is a pity as it is one of the unique advantages of quadrupole ICP-MS to combine dating and concentration determination (for the future. . .).

AGREE. In retrospect, additional chemistry would have been a useful technique to help discriminate between the two biotite phases. Unfortunately, this was not done. For the biotite chemistry, see above point. For muscovite and phengite, the differentiation was purely petrographic, with muscovite being in larger sheets (e.g., Fig. 3h) and phengite being microcrystalline (Fig. 3f). We have stated this now in various places in the results section.

4.) How many age domains? Are there really "only" two significant ages extractable from the biotite data? A close inspection of figure 4b and 4d seems to show a clear linear trend of your "biotite 1" that falls on an isochron somewhere between the 2.5 Ga and 1.2 Ga. It could well be in this 1.8-1.6 Ga interval mentioned in chapter 2.1.    I strongly recommend replotting the data from figure 4 in a histogram where x-axis is single spot model ages (like kernel density plots typical for detrital zircons). With a bit of statistics it can be tested if this third age is significant.   For me it looks to regular    to be a product of partial resetting (in contrast,  the scatter visible by the grey circles   in Figure 4a is a convincing example of partial resetting). IF such a third age can be established, the next question is: what distinguishes "biotite1" from samples in fig 4b and d from biotite from samples in fig 4a?

AGREE. As the above points, these dates younger than 2.5 Ga are physical mixtures and geologically meaningless.

5.)   Closure vs formation ages:  as far as I can see,  the difference between closure   and formation ages has been handled in a succinct matter. Still, I think the manuscript will benefit by giving this topic a bit more prominence (e.g.,  defining those topics in  the introduction and devoting a chapter in the discussion). It is currently a hot topic ("petrochronology" vs "thermochronology"), not always trivial to say which process is dominating in specific cases. Here you have (at least) two age populations of biotite  not governed by volume diffusion (still, it is worth mentioning grain size effects on closure temperatures). Furthermore, muscovite is supposed to have higher closure temperatures than biotite, yet muscovite is younger than biotite 1, a clear indication that muscovite ages are formation ages.

PARTLY AGREE. The muscovite ages have high uncertainties (+/- 170 Ma) that overlap with biotite 1; thus, it is not possible to say that the biotite 1 is younger than the muscovite. Thus, we find it difficult to comment on the closure temperature and formation ages.

Specific edits:
-        abstract, line 16: instead of "K- and Rb-bearing", say "K-rich" or "Rb-bearing".

AGREE. Changed.

-        abstract, line 20: replace "second assemblage" with "younger assemblage" (second and first are not defined in abstract)

AGREE. Changed.

-        introduction, line 47: papers by Wolfgang Muller are good examples for texturally-controlled micromilling Rb-Sr dating (e.g., EPSL 180, 385-397).

AGREE. This paper has been cited.

-        methods, line 193: instead "zero counts" (which do not exist), rather say "< xx cps".

AGREE. We have changed this to <7 cps.

- methods, line 217: please say a few more words on the biotite secondary standard CK001B. What chemistry does it have (annite vs phlogopite, etc)? Micas are a very large group with very different chemistry- it may turn out that "matrix" effects may even operate within a mica group, so it is good to have the chemistry specified.

We currently have another paper under review that specifically characterises this secondary standard.

- results, line 237-243: out of curiosity: do you have any constraints on the temperature conditions during mylonite formation? Would be nice to state if you can, as it will give important constraints that biotite 1 can statically survive a certain heating episode without diffusional resetting.

AGREE. Unfortunately, we have no calculation of temperature based on petrology. We believe that a mineralogy of the Assemblage 2 indicates ingress of fluids during the shearing and demonstrates an open thermodynamic system, which makes any PT calculations unreliable. Textural evidence shows brittle deformation of K-feldspar supporting greenschist facies temperatures.

- results, line 271: "fractures were intercepted". Again, this is not clearly visible in the petrographic image. A BSE or reflected light image would be better.

AGREE. Following the comment above, we have taken selected BSE images to demonstrate this.

- discussion, line 313-316: please delete the last two sentences, as they are too speculative!! You can ask Steve Reddy how messy cite occupancy of radiogenic isotopes are on an atomic level. Unless somebody is doing AFM on a mica, we simply do not know.

AGREE. We have deleted these.

- figure 1b: please note that yellow diamonds are gold occurrences. Best wishes,

AGREE. This has been added to the legend map.

[revised manuscript text omitted]